# Electrochemical reduction of acetonitrile to ethylamine

Rong Xia [1,2], Dong Tian[3], Shyam Kattel [4], Bjorn Hasa [1], Haeun Shin [1], Xinbin Ma [2✉], Jingguang G. Chen [3✉] & Feng Jiao [1✉]

Electrifying chemical manufacturing using renewable energy is an attractive approach to reduce the dependence on fossil energy sources in chemical industries. Primary amines are important organic building blocks; however, the synthesis is often hindered by the poor selectivity because of the formation of secondary and tertiary amine byproducts. Herein, we report an electrocatalytic route to produce ethylamine selectively through an electroreduction of acetonitrile at ambient temperature and pressure. Among all the electrocatalysts, Cu nanoparticles exhibit the highest ethylamine Faradaic efficiency (~96%) at −0.29 V versus reversible hydrogen electrode. Under optimal conditions, we achieve an ethylamine partial current density of 846 mA cm$^{-2}$. A 20-hour stable performance is demonstrated on Cu at 100 mA cm$^{-2}$ with an 86% ethylamine Faradaic efficiency. Moreover, the reaction mechanism is investigated by computational study, which suggests the high ethylamine selectivity on Cu is due to the moderate binding affinity for the reaction intermediates.

[1] Center for Catalytic Science and Technology, Department of Chemical and Biomolecular Engineering, University of Delaware, Newark, DE, United States. [2] Key Laboratory for Green Chemical Technology of Ministry of Education, Collaborative Innovation Center of Chemical Science and Engineering, School of Chemical Engineering and Technology, Tianjin University, Tianjin, China. [3] Department of Chemical Engineering, Columbia University, New York, NY, United States. [4] Department of Physics, Florida A&M University, Tallahassee, FL, USA. ✉email: xbma@tju.edu.cn; jgchen@columbia.edu; jiao@udel.edu

Amines are essential building blocks and intermediates for numerous pharmaceuticals, agrochemicals, polymers, dyestuffs, emulsifiers, and fine chemical products[1–4]. Ethylamine alone has an annual global market of $1.85 \times 10^5$ tons[5]. There are several approaches for ethylamine production, such as the amination of alcohols[6,7], reductive amination of aldehydes or ketones[8,9], and hydrogenation of nitriles[10,11]. Among those methods, direct hydrogenation to ethylamine has drawn special interest due to its unique acetonitrile raw material source. Acetonitrile is mainly produced as a byproduct in acrylonitrile production through the Sohio process[5,12]. Currently, only a small fraction of acrylonitrile producers recover acetonitrile and most of the acetonitrile is burned as fuels, which emits a significant amount of $NO_x$[5]. Acetonitrile reduction to ethylamine could provide a more environmentally friendly process converting excess acetonitrile manufacturing capacity to value-added ethylamine product. During acetonitrile hydrogenation, the primary ethylamine attacks the imine intermediates to form secondary and tertiary amines because of the nucleophilic property of amine. As a result, the product is often a mixture of ethylamine, diethylamine, and triethylamine with a low selectivity of the primary ethylamine[5]. For this reason, past efforts have been devoted to developing metallic catalysts, such as Ni[13], Cu[14], Pt[15], Pd[16], and Sn[17], exhibiting an enhanced ethylamine selectivity in thermocatalytic hydrogenation of acetonitrile. However, the improved ethylamine selectivity was achieved using expensive strategies, such as strong acid trapping the primary amine[18], an excessive amount of ammonia shifting the equilibrium[19], and elevating the hydrogen partial pressure substantially.

As the ever-increasing deployment of renewable energy production substantially reduces the electricity cost over the past decades, cheap, renewable electricity provides new opportunities in electrifying chemical transformations as a potential route to decarbonize the chemical industries[20–24]. Electrochemical reduction of acetonitrile provides an alternative route and could potentially overcome the selectivity issues of thermal catalytic hydrogenation. The reaction schemes for nitrile electroreduction are as follows:

$$CH_3CN + 4H_2O + 4e^- \rightarrow CH_3CH_2NH_2 + 4OH^- \text{ (cathode)}$$
(1)

$$4OH^- \rightarrow 2H_2O + O_2 + 4e^- \text{ (anode)}$$
(2)

As illustrated in Fig. 1, electrochemistry driven by renewable energy uses water as hydrogen sources and is carried out under ambient conditions that could potentially convert the acetonitrile to ~100% ethylamine. Activation of acetonitrile electrochemically has been reported in the literature. Previous studies showed that acetonitrile can be converted to 3-aminocrotonitrile anion on a platinum electrode in a non-aqueous electrolyte[25,26]. In addition, the chemisorption of acetonitrile on Pt electrode coupled with two consecutive electron and proton transfer processes was qualitatively investigated using cyclic voltammetry, in situ infrared spectroscopy, and online mass spectrometry[27–31]. More recently, Child et al., utilized homogeneous cobalt-based molecular catalysts to electrochemically reduce acetonitrile to ethylamine using acetic acid as a hydrogen source in organic electrolyte[32]. Despite these efforts, electrochemical reduction of acetonitrile suffers from low Faradaic efficiency (FE, ~22%) and low current density (~0.04 mA cm$^{-2}$) and has not been systematically investigated using metallic electrocatalysts.

Here, we report an electrochemical approach for acetonitrile reduction. In order to gain fundamental knowledge of acetonitrile electrochemical reduction, seven metal catalysts are initially screened in a flow cell electrolyzer to determine the optimal catalysts. The maximum Faradaic efficiency (FE) of ~95% for ethylamine is achieved at −0.29 V versus reversible hydrogen electrode (RHE, all potentials reported in this paper are converted to the RHE scale) on Cu nanoparticles catalysts. The mass transportation of reactants and the impact of pH on acetonitrile electroreduction are studied to improve the performance at high current density (~1 A cm$^{-2}$). The online electrochemical differential flow electrolyzer mass spectrometry (FEMS) is employed for the time-resolved detection of the product and density functional theory (DFT) calculation is applied for further understanding of the reaction mechanism.

## Results

**Catalyst screening and performance analysis.** The initial screening of electrocatalysts for acetonitrile reduction was conducted using seven monometallic catalysts, i.e., Cu, Ni, Pd, Pt, Bi, In, and Sn, which are commonly used in the thermocatalytic nitrile hydrogenation reaction. The acetonitrile reduction activity was evaluated in a two-compartment microfluidic flow cell. Catalysts loaded on porous carbon paper served as a cathode and Ni foam was used as an anode. Nafion 211 membrane was placed between the cathodic and anodic chambers. A 1 M NaOH aqueous solution was fed as anolyte and a 1 M NaOH solution containing a specific concentration of acetonitrile was used as catholyte. The performance was analyzed at different applied potentials ranging from −0.4 V to −0.65 V in 8 wt% acetonitrile in 1 M NaOH electrolyte. The maximum ethylamine FE together with the corresponding H$_2$ FE of each catalyst is summarized in Fig. 2a (more details in Supplementary Fig. 1). The gas product (H$_2$) was analyzed by gas chromatography and the amount of ethylamine product was quantified by $^1$H nuclear magnetic resonance (NMR) spectroscopy. Among all the monometallic catalysts, Cu showed the highest ethylamine FE of 94.6%, with a total current density of 50 mA cm$^{-2}$. In sharp contrast, Pt, Sn, In, and Bi showed very low ethylamine FE (<3%), and the product was dominated by H$_2$. At industrially relevant current densities (>150 mA cm$^{-2}$), the Cu catalyst maintained a decent ethylamine FE > 72%, whereas the ethylamine FEs for Ni and Pd dropped to less than 20%. At the optimal conditions, the maximum production rate of ethylamine on Cu was 3.135 mmol cm$^{-2}$ h$^{-1}$ (Supplementary Fig. 3), which was approximately three times as Ni and Pd, and 31 times as Pt. Another important observation is that neither diethylamine nor triethylamine was detected over all the catalysts by NMR, although they are major byproducts in the thermocatalytic nitrile hydrogenation reaction, suggesting that electroreduction of acetonitrile is a highly selective approach to produce primary amine from its corresponding nitrile. The improved ethylamine selectivity in electrocatalytic acetonitrile reduction over thermocatalytic hydrogenation routes is likely due to its mild reaction temperature because high reaction temperatures promote secondary amine formation[33,34]. We further extended our investigations to three different Cu catalysts, i.e., Cu nanoparticles (25 nm), Cu microparticles (0.5–1.5 μm), and oxide-derived Cu. The Cu nanoparticles exhibited the best performance due to high surface area (see Supplementary Figs. 4–8 for more detailed description). At a total current density of 1 A cm$^{-2}$, Cu nanoparticles showed a maximum ethylamine partial current density of 557 mA cm$^{-2}$ (with an ethylamine FE of 55.7%) at −0.76 V (Fig. 2b).

**Influence of acetonitrile concentration and pH value on acetonitrile electroreduction.** The dependence of acetonitrile concentration was investigated by varying the mass fraction of acetonitrile in electrolytes. The ethylamine FE was observed to increase with the concentration of acetonitrile from 4 wt% to 12 wt% (Supplementary Fig. 8). The ethylamine partial current density achieved 846 mA cm$^{-2}$ at −0.73 V in 12 wt% acetonitrile

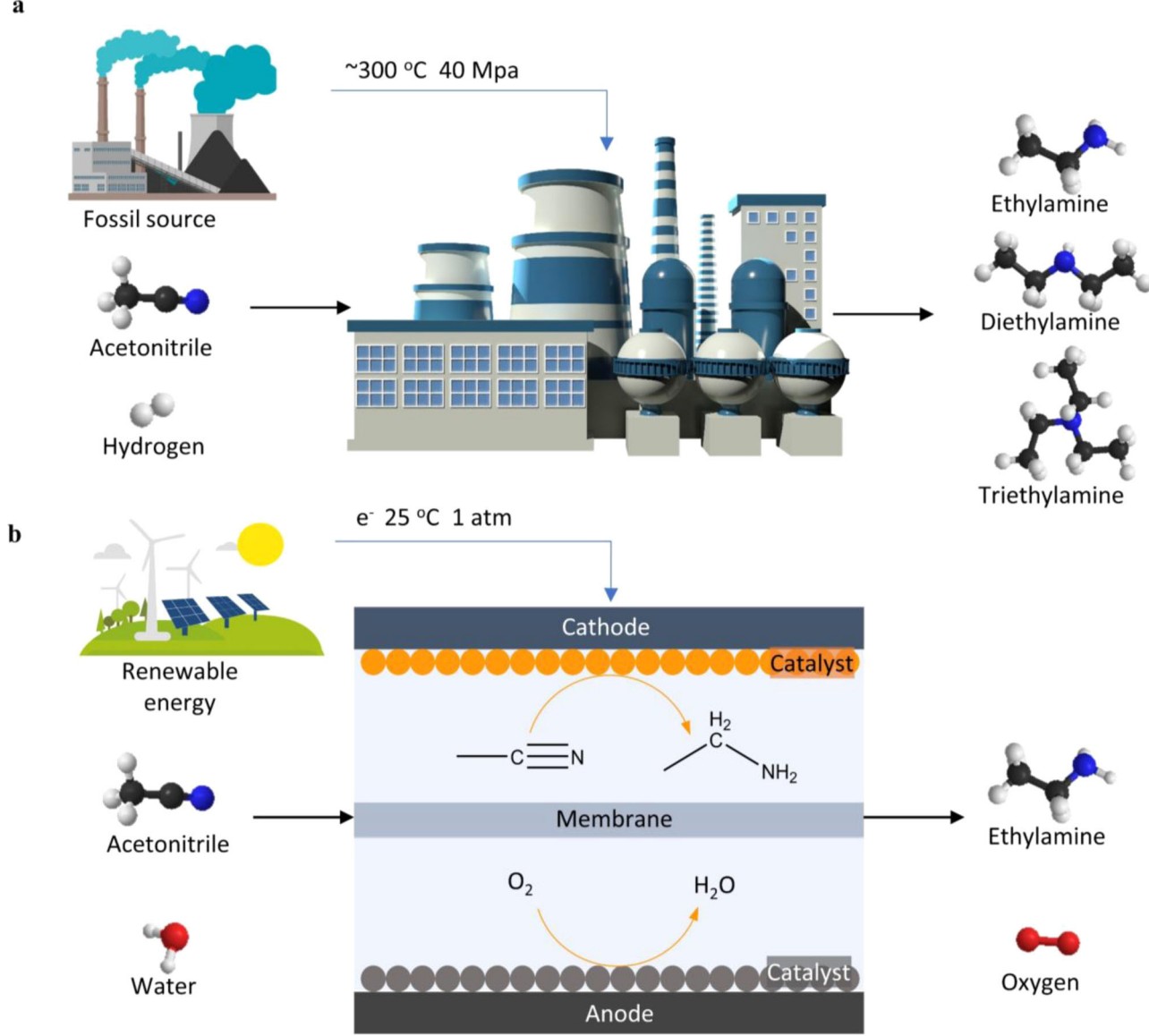

**Fig. 1 Schematic comparison of acetonitrile reduction to ethylamine. a** Thermal catalytic hydrogenation of acetonitrile with a relatively low selectivity towards primary amine, and (**b**) electroreduction of acetonitrile with a >90% selectivity towards primary amine.

(Fig. 3a), which was 1.8 times as in 8 wt% acetonitrile and 3.0 times as in 4 wt% acetonitrile. The near exponential increase of ethylamine partial current density was independent of applied potentials in 12 wt% acetonitrile, suggesting that the acetonitrile mass transportation limitation to the electrode interface was minimal. The reaction order was determined with respect to the concentration of acetonitrile. The ethylamine partial current density ($j_{ethylamine}$) was measured at −0.45 V in 1 M NaOH containing various concentrations of acetonitrile. The reaction order (n) was determined as:

$$j_{ethylamine} = const\,[CH_3CN]^n\,exp(-FE/RT) \qquad (3)$$

which was corresponding to the slope of log $j_{ethylamine}$ versus log [$CH_3CN$]. As shown in Supplementary Fig. 9, the reaction order with respect to acetonitrile concentration is about 0.91, indicating an approximate first-order dependence on acetonitrile concentration.

To elucidate the pH effect on the acetonitrile electroreduction, we investigated various aqueous electrolytes with different pH values and the results are summarized in Fig. 3b and Supplementary Fig. 9. In an acid electrolyte (i.e., 0.5 M $H_2SO_4$), hydrogen evolution

reaction (HER) was the dominant reaction and acetonitrile reduction was largely suppressed. Interestingly, the acetonitrile reduction performance in 0.5 M $Na_2SO_4$ was even worse than 0.5 M $H_2SO_4$. In contrast, 1 M NaOH exhibited a significantly better performance in acetonitrile electroreduction than neutral and acidic electrolytes. Increasing the concentration of NaOH to 2 M further improved the ethylamine partial current density to 635 mA cm$^{-2}$ (at −0.73 V), suggesting that a high pH was preferred for acetonitrile electroreduction. A similar pH effect was also observed in $CO_2$ electroreduction, where alkaline was the most effective electrolyte to suppress undesired HER[35,36]. The pH effect was further investigated by tuning the concentration of NaOH from 0.1 to 1 M while keeping the total Na$^+$ concentration as 1 M (with the addition of $Na_2SO_4$) in all studies, which allowed us to exclude the potential impact of Na$^+$. The results (Fig. 3b) showed that the ethylamine partial current density increased with the basicity of electrolyte solution across the applied potentials. The ethylamine partial current densities were also plotted against the potentials relative to the standard hydrogen electrode in Supplementary Fig. 11, which shows pH-independence of the ethylamine formation, suggesting that the rate-determining

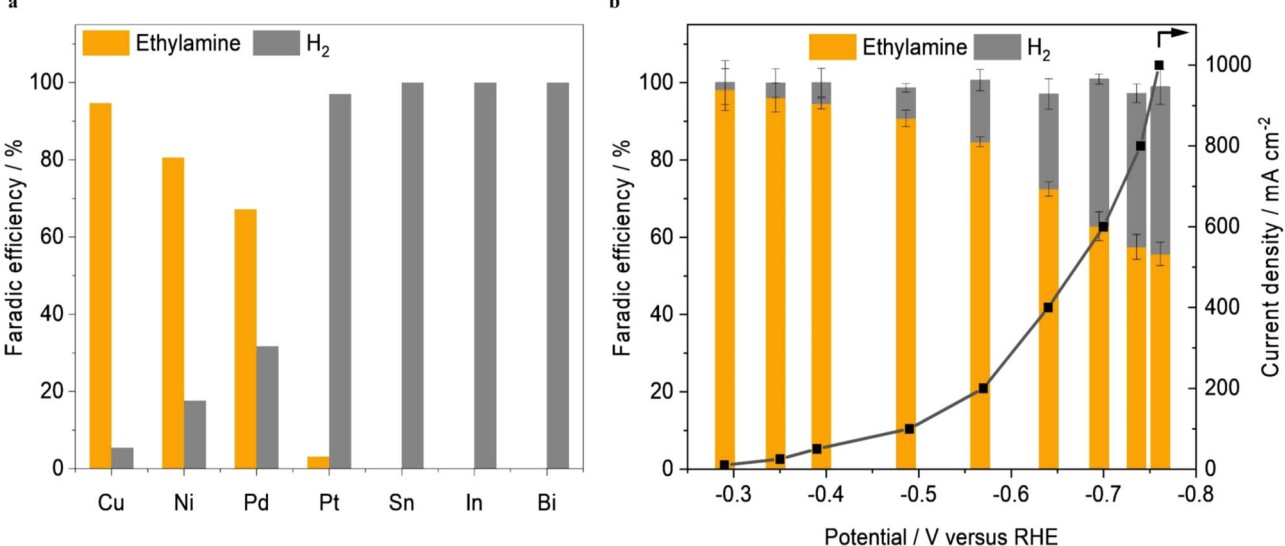

**Fig. 2 Comparison of acetonitrile reduction activity on different catalysts. a** The highest FE of ethylamine on various metal catalysts and corresponding $H_2$ FE in the applied potential range of −0.4 V to −0.65 V versus RHE. **b** Ethylamine FE, $H_2$ FE, and corresponding current density versus applied potentials on Cu nanoparticles in acetonitrile reduction (8 wt% acetonitrile in 1 M NaOH as electrolyte). Error bars represent the standard deviation from at least three independent measurements.

step of acetonitrile electroreduction in alkaline conditions does not involve hydroxide as one of the reactants. We speculate that the initial activation of acetonitrile (either single-electron reduction of –CN or proton-coupled electron transfer) could be the rate-determining step and $H_2O$ is likely the proton source.

The stability of Cu nanoparticle catalyst in acetonitrile electroreduction was also studied using a flow cell at a constant current density of 100 mA cm$^{-2}$ for 20 h in 1 M NaOH electrolyte containing 8 wt% acetonitrile. Figure 3c shows that both a stable potential (~−0.46 V) and a steady ethylamine FE (>86%) were achieved over the 20-h stability test. The slight potential disturbance was mainly caused by the gas bubbles ($H_2$) periodically accumulated and flushed out by the liquid electrolyte. The structural stability of the Cu nanoparticles was examined by scanning electron microscopy (SEM) and X-ray photoelectron spectroscopy (XPS). No obvious changes were observed (Supplementary Fig. 12), although the post-reaction electrode showed a loss of catalyst particles from the porous carbon paper, which may contribute to a slight decrease of ethylamine FE in the stability test.

**Product distribution of acetonitrile electroreduction**. We employed in situ FEMS to probe the reaction intermediates/products distribution during the electrolysis. FEMS offered a time-resolved detection with high sensitivity at ~10–100 μm from the electrode surface[37]. A schematic diagram of the FEMS configuration is shown in Supplementary Fig. 13. A scan of fragments with mass to electron ratio (m/z) ranging from 1 to 90 was initially performed at −0.5 V versus RHE (~100 mA cm$^{-2}$). The background-subtracted spectrum (Fig. 4a) showed mass fragments corresponded to water, acetonitrile, hydrogen, and ethylamine. It is shown that a traceable amount of diethylamine was formed during acetonitrile electrochemical reduction and no triethylamine was detected on the electrode surface, while they are two major byproducts in thermal acetonitrile hydrogenation (see Supplementary Fig. 14 for more detailed description). This result confirmed the high selectivity of ethylamine in acetonitrile electroreduction. We further examined the onset potential of each product using FEMS coupled with linear sweep voltammetry (0 to −0.76 V, the scan rate of −5 mV s$^{-1}$) in 1 M NaOH containing

8 wt% acetonitrile. The fragment intensities recorded in the mass spectrometer were synchronized with the potentiostat signal (Fig. 4b). Hydrogen, ethylamine, diethylamine, and triethylamine signals were recorded at m/z 2, 45, 58, and 86, respectively. As shown in Fig. 4b, the onset potential of hydrogen evolution was about −0.21 V versus RHE, ethylamine started at −0.23 V, diethylamine signal emerged at −0.32 V and no change of triethylamine signal was observed. The FEMS results clearly show that ethylamine was formed at a lower overpotential than diethylamine, which is in good agreement with the hypothesis that diethylamine evolved from the condensation reaction between ethylamine and imine intermediate.

**DFT calculations**. DFT calculations were performed to rationalize the experimentally observed activity trends of acetonitrile reduction reaction (nitrile reduction hereafter) over various metal catalysts. To this end, spin-polarized periodic DFT calculations were carried out to compute the binding energies of reaction intermediates involved in the nitrile reduction reaction on Cu(111), Ni(111), and Pt(111). Optimized geometries in Supplementary Fig. 14 show that the intermediates bind via unsaturated N and or C atoms mostly at the surface top and hollow sites. It is observed (Supplementary Table S1) that the binding strength follows the order Ni(111) > Pt(111) > Cu(111) for the intermediates that solely bind via N and Pt(111) > Ni(111) > Cu(111) for the intermediates that bind via hydrogenated N (i.e., $NH_x$) and or C/$CH_x$. The DFT calculated binding energies were used to calculate the free energy changes for the nitrile reduction reaction along four potential pathways shown in Table S2 using the computational hydrogen electrode model[38]. DFT-calculated free energy diagrams in Supplementary Figs. 15 and 16 show that $CH_3CN$ reduction to $CH_3CH_2NH_2$ most likely occurs via pathways 2, 1, and 3 (among four possible pathways listed in Supplementary Table S2) on Cu (111), Ni(111), and Pt(111), respectively. A comparison of free energy diagrams calculated at an applied potential $U = 0$ V along the most favorable pathways in Fig. 5 demonstrates that nitrile reduction is thermodynamically more favorable on Cu(111) compared to Ni(111) and Pt(111). The relatively milder binding affinity of Cu(111) for the *$CH_3C_xN_y$ intermediates makes the entire process facile, with the most difficult step (adsorption of

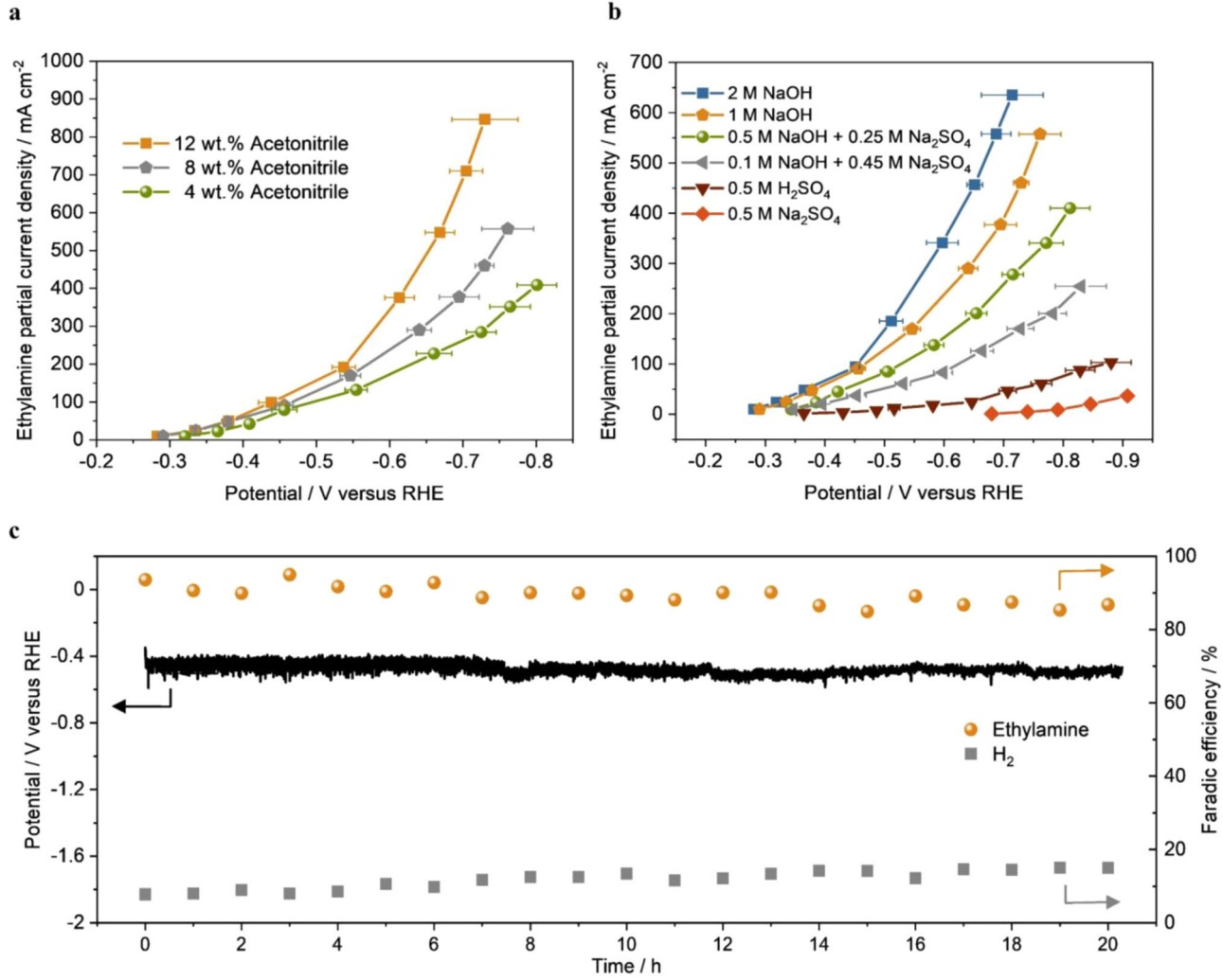

**Fig. 3 Acetonitrile concentration and pH value effect on the electroreduction of acetonitrile and stability test. a** Ethylamine partial current density under different concentrations of acetonitrile. **b** Ethylamine partial current density under electrolyte of different pH values. **c** Stability test over a span of 20 h using Cu nanoparticles as the catalyst at a constant current density of 100 mA cm$^{-2}$ (8 wt% acetonitrile in 1 M NaOH as electrolyte). Error bars represent the standard deviation from at least three independent measurements.

*CH$_3$CN) being uphill in energy by 0.20 eV. In contrast, the most difficult steps on Ni(111) [*CH$_3$CH$_2$N reduction] and Pt(111) [desorption of *CH$_3$CH$_2$NH$_2$] are uphill in energy by 0.26 eV and 0.45 eV, respectively. Thus, the DFT results predict that the activity for CH$_3$CN reduction to CH$_3$CH$_2$NH$_2$ should follow the order: Cu(111) > Ni(111) > Pt(111). As in many electrochemical reduction reactions, HER is a competing reaction in the nitrile reduction reaction. The DFT-calculated free energy diagrams (Supplementary Fig. 17) predict the HER activity in the order Pt (111) > Ni(111) > Cu(111). Importantly, free energy diagrams show that the nitrile reduction reaction is energetically more favorable than the HER on Cu(111) and Ni(111) (by 0. 16 eV and 0.08 eV, respectively). In contrast, HER is found to be more favorable (by 0.19 eV) than the nitrile reduction on Pt(111). Thus, these DFT-calculated free energy changes of the rate-limiting steps in acetonitrile electroreduction and HER are consistent with the experimentally measured trend of ethylamine FE on Cu, Ni, and Pt (Cu > Ni > Pt) (Fig. 2a). Overall, the DFT results are in excellent agreement with the experimental observations and suggest that Cu mainly promotes acetonitrile electroreduction due to moderate binding affinity for the reaction intermediates. In comparison, Pt selectively promotes the HER while the acetonitrile electroreduction and HER activity of Ni lie in between Cu and Pt.

In this study, we conducted a systematic investigation of acetonitrile electroreduction to primary amine (ethylamine) under ambient conditions in a flow cell. Compared with Pd, Pt, Ni, Bi, Sn, and In, Cu showed the highest performance and an ethylamine FE of 96% was obtained at −0.29 V. Further studies showed that alkaline conditions were the most favorable for the selective ethylamine production and suppressed the competing HER on the cathode. At the optimal conditions, an ethylamine partial current density of 846 mA cm$^{-2}$ was achieved using a Cu nanoparticle catalyst at −0.73 V in a 1 M NaOH electrolyte containing 12 wt% acetonitrile. Finally, the reaction mechanism of acetonitrile electroreduction on Cu catalysts was also investigated using FEMS and DFT calculations. This work provides an alternative route for primary amine production, which overcomes the challenges associated with poor selectivity in thermocatalytic nitrile hydrogenation and paves the way towards electrification of chemical manufacturing.

## Methods
**Preparation of electrodes**. Cu nanoparticles (25 nm, 99.99%), Cu microparticles (0.5–1.5 μm) were purchased from Sigma Aldrich. Oxide-derived Cu (OD-Cu) was prepared through electrochemical deposition of Cu$_2$O film on porous carbon paper (see details in the Supporting information). The as-deposited Cu$_2$O was reduced at

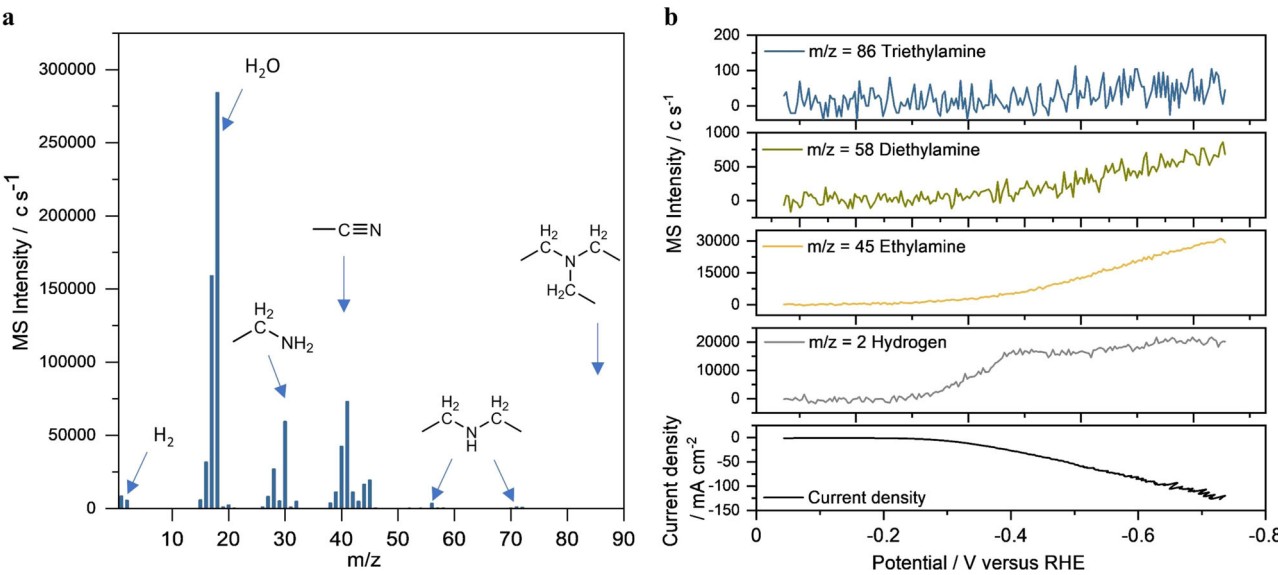

**Fig. 4 The product distribution analysis by flow electrolyzer mass spectroscopy (FEMS). a** Mass spectrometric signals recorded while scanning in a mass range of 0−90 amu at −0.5 V versus RHE. **b** Linear sweep voltammetry measurement ranging from 0 V to −0.76 V versus RHE (scan rate of −5 mV s⁻¹). The potentiostat signal (black line) was synchronized with the fragment intensity of hydrogen (gray line), ethylamine (orange line), diethylamine (green line), and triethylamine (blue line) recorded in a mass spectrometer. The m/z stands for mass-to-charge ratio.

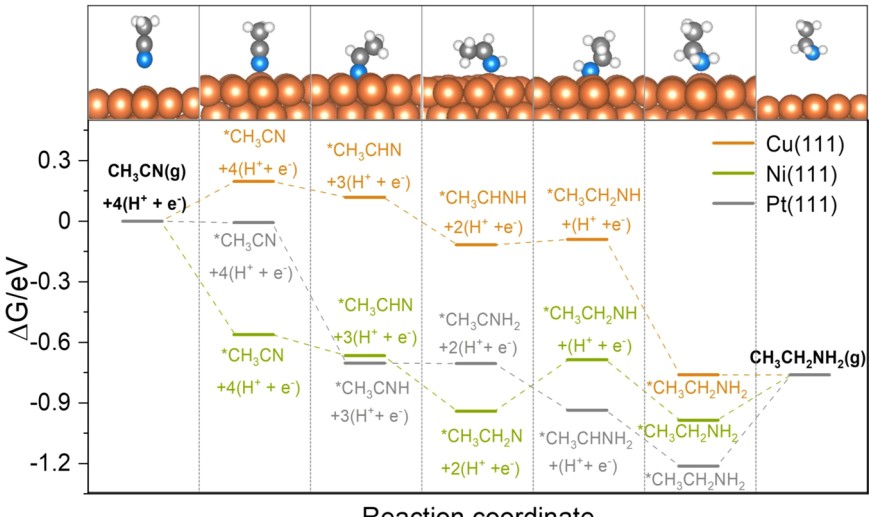

**Fig. 5 DFT calculations.** Free energy diagrams for $CH_3CN(g)$ electroreduction to $CH_3CH_2NH_2(g)$ along the most favorable pathway on Cu(111), Ni(111), and Pt(111) at an applied potential $U = 0$ V.

10 mA cm⁻² in 1 M NaOH for 15 min to form OD-Cu. Ag (20 nm, 99.99%), Bi (80 nm, 99%), and In (80 nm, 99%) were purchased from US Research Nanomaterials, Inc. Sn (<150 nm, 99.99%), Pd/C (5 wt%, 99.99% metal basis), and Pt/C (5 wt%, 99.99% metal basis) was purchased from Alfa Aesar. For a typical Cu nanoparticles electrode, 25 mg Cu nanoparticles were dispersed in 3 ml isopropanol, and 20 μl Nafion ionomer (5 wt% in H₂O) was added as a binder. The catalyst ink was sonicated for 20 min and drop-casted onto a 2.5 cm² porous carbon paper (Sigracet 39BC, Fuel Cell Store). The loading was controlled at 0.5 mg cm⁻² based on the metal mass. A similar procedure was applied to other metal catalysts.

**Material characterization**. SEM image and corresponding EDX elemental mapping were taken using an Auriga 60 Cross Beam SEM. The surface composition of the electrode was determined using a K-Alpha X-ray photoelectron spectrometer system (Thermo Fisher Scientific). The XPS data were analyzed using CasaXPS and the adventitious C 1s signal was calibrated to 284.5 eV. The electrochemical surface area (ECSA) was determined by measuring the double-layer capacitance ($C_{dl}$) in Ar-saturated 0.1 M HClO₄ solution. The ECSA is given by:

$$ECSA = R_f S \qquad (4)$$

Where $R_f$ is the roughness factor normalized by the Cu foil and $S$ stands for the ideal surface area of smooth Cu foil electrode (1 cm²). The roughness factor was estimated by normalizing the double-layer capacitance $C_{dl}$ to that of a Cu foil. The $C_{dl}$ was determined by measuring the capacitive current in the non-faradaic potential region under various scan rates of cyclic voltammetry(10 mV s⁻¹, 20 mV s⁻¹, 40 mV s⁻¹, 60 mV s⁻¹, 80 mV s⁻¹and 100 mV s⁻¹). The $C_{dl}$ was obtained by plotting the capacitive current against the scan rates.

**Flow cell electrocatalysis**. Acetonitrile electroreduction was performed in a two-compartment microfluidic flow cell electrolyzer. The overall reaction is composed of two half-cell reactions: acetonitrile reduction on the cathodic side and oxygen evolution reaction on the anodic side. The Nafion 211 membrane (Fuel Cell Store) isolated the two chambers. Nafion 211 membrane was utilized in this study for its relatively high stability in the presence of organics. In a typical acetonitrile electrochemical reduction, the catholyte was 8 wt% acetonitrile (99.8%, Sigma-Aldrich) 1 M NaOH (99.99%, Sigma-Aldrich) aqueous solution, and the anolyte was 1 M NaOH aqueous solution. The flow rates of electrolytes were imposed by two peristaltic pumps at designed values. The catholyte went through the electrolyzer and the effluent was connected to a gas–liquid separator to separate the gas product

with liquid. Aragon purged the gas in the headspace of gas–liquid separator to the gas chromatography (SRI Instruments) and the outlet flow rate of GC was measured by the flow meter (Agilent ADM).

Chronopotentiometry experiments were conducted to evaluate the acetonitrile electroreduction performance using an Autolab PG128N potentiostat. A three-electrode set-up was applied, and the Ag/AgCl electrode (Pine Research) was used as the reference electrode. Nickel foam was used as the anode for all the experiments in the basic electrolytes and IrO$_2$ deposited on Ti felt was used in an acid environment. The current-interrupt technique was used to measure the resistance between the working electrode and the reference electrode before the potential was applied. The measured potentials were compensated by the iR correction for the voltage drop caused by the solution resistance and were converted into the RHE as follows:

$$E(\text{versus RHE}) = E(\text{versus Ag/AgCl}) + 0.210\,\text{V} + 0.0591 \times \text{pH} + iR \quad (5)$$

For each data point, the electrolysis was carried out for 600 s to reach a steady-state before injected into the GC, and then the liquid product was collected for 300 s.

The gas product was quantified via a gas chromatography system installed with HayeSep D and Mol Sieve 5 A columns that connected to a thermal conductivity detector and a flame ionization detector. Argon (99.999%) was applied as the carrier gas. Liquid products were identified through a Bruker AVIII 600 MHz NMR spectrometer. In short, 500 μl of the sampled catholyte was added with 100 μl of internal standard solution that consisted of 25 ppm (v/v) dimethyl sulfoxide (≥99.9% (Alfa Aesar)) in D$_2$O. Presaturation method water suppression was applied to analyze the one-dimensional $^1$H spectrum.

**Flow electrolyzer mass spectrometry**. The determination of gaseous and volatile reaction products was carried out by online mass spectra in parallel to the electrochemical measurement. The reaction products were collected through a PEEK capillary (McMaster, inner diameter 0.25 mm) covered by a hydrophobic PTFE membrane (TISCH, pore size 200 μm). The membrane prevents the introduction of an aqueous electrolyte while allowing the volatile and gas products to enter the vacuum chamber. The capillary was in contact with the cathode surface as is shown in Supplementary Fig. 13. DUO 20 M (Pfeiffer) and Hidden Quadrupole were used as differential pumps and mass spectrometer, respectively. Product samples were ionized at an ionization potential of 70 eV using a secondary electron detection voltage of 1700 V with an emission current of 200 μA.

**Computational methods**. DFT calculations utilized spin polarization (DFT)[39,40] at the GGA level within the PAW-PW91 formalism[41,42] using the Vienna ab initio simulation package (VASP) code[43,44]. A $3 \times 3 \times 1$ Monkhorst-Pack grid[45] was adopted for the Brillion zone integration and a plane wave cut-off energy of 400 eV was used for the total energy calculations

Low index (111) surfaces of Pt, Ni, and Cu were modeled using four-layer $3 \times 3$ surface slabs. In order to minimize the artificial interactions between the surface and its periodic images, a vacuum layer of ~15 Å thick was added in the slab cell along the direction perpendicular to the surface. Atoms in the bottom two layers were fixed while all other atoms were allowed to relax until the Hellman–Feynman force on each ion was smaller than 0.01 eV/Å. The binding energy (BE) of adsorbate was calculated as:

$$\text{BE}(\text{adsorbate}) = E(\text{slab} + \text{adsorbate}) - E(\text{slab}) - E(\text{adsorbate}) \quad (6)$$

where $E(\text{slab} + \text{adsorbate})$, $E(\text{slab})$, and $E(\text{adsorbate})$ are the total energy of slab with adsorbate, clean slab, and adsorbate in the gas phase, respectively. For the calculation of hydrogen binding energy, $E(H)$ is taken as one-half of the total energy of the H$_2$ molecule. The Gibbs free energy $(G)$ is calculated as[38]

$$G = E + \text{ZPE} - TS \quad (7)$$

where $E$ is the total energy obtained from DFT calculations, and ZPE and $S$ are the zero-point energy and entropy of a species, respectively, at $T = 298$ K.

## Data availability
The data that support the findings of this study are available from the authors on reasonable request, see author contributions for specific data sets.

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

## Acknowledgements

R.X. would like to acknowledge the financial support from the China Scholarship Council (CSC). The authors at Tianjin University thank the financial support from the National Natural Science Foundation of China (21325626) and Tianjin Key Science and Technology Project (19ZXNCGX00030). The authors at Columbia University acknowledge support from the US Department of Energy, Office of Basic Energy Sciences, Catalysis Science Program (Grant no. DE-FG02- 13ER16381). DFT calculations were performed using resources of the Center for Functional Nanomaterials, which is a U.S. Department of Energy (DOE) Office of Science Facility, and the Scientific Data and Computing Center, a component of the BNL Computational Science Initiative, at Brookhaven National Laboratory under Contract no. DE-SC0012704.

## Author contributions

F.J., J.G.C., and X.M. supervised the project. R.X. designed the experiments and conducted the electrochemical measurements. D.T. and S.K. conducted the DFT calculation. B.H. performed the FEMS study. H.S. conducted the SEM measurements. All authors discussed the results and contributed to manuscript preparation.

## Competing interests

The authors declare no competing interests.
