## [Peer Review File · Nature Communications]

REVIEWER COMMENTS

Reviewer #1 (Remarks to the Author):

This article presents a good and potentially highly applicable results, the authors are absolutely correct that using electrochemistry to produce organic chemicals and feedstocks could be a great way to use at times excessive solar, wind, and other renewable energy. The study is well executed at a good level, shows a good understanding of the chemistry in question and is potentially very high impact.

However, they need to review and cite the literature more carefully before this can be published. In what way do they mean it is the "first-of-its-kind study of electrochemical reduction of acetonitrile to ethylamine"? Because there is definitely some prior art. For example, Dalton Trans 2019, 48, 9576 presents an example with a molecular cobalt catalyst - much less promising in FE and stability than the current work, but nevertheless a study of the process in question. And there is older work based on solid-state metal catalysts - see for example Batanero et al, J Org Chem 2002, 67, 2369; Foley et al, Can J Chem 1988, 66, 201 - and I think there are other examples going further back than this.

In summary, this is potentially publishable in Nature Comms but it needs to properly acknowledge prior work, and in the context of that work - largely published in much lower impact journals - make a convincing case for the results and insights obtained.

Reviewer #2 (Remarks to the Author):

The manuscript describes the electrochemical reduction of acetonitrile to ethylamine. The authors correctly explain that there is a need to explore the application of electrochemical methods for the synthesis of commodity and/or high valuable products. The manuscript includes a very systematic study of catalyst and reaction parameters as well as computational studies to provide information about the reaction mechanism of acetonitrile electroreduction. The topic and this study are of high interest to the scientific community and can be published in Nature Communications. However, before publication the authors should clarify and/or adjust the following points:

- 1- Can the authors elaborate on the source (and production process) of acetonitrile used as starting material for the synthesis of ethylamine? The sustainability of the production process is highly dependent on the source of the substrate and acetonitrile has problems of itself. It is of high importance to have clarity when suggesting alternative synthetic routes.
- 2- The redox equations for the electrochemical process should be provided as they help not expert reader to better understand the proposed process.
- 3- The authors used a gas diffusion layer as support for the catalyst used. However, the reaction does not involve gas reactants. In this case the GDL, becomes a simple carbon-based support and the use of GDL nomenclature can lead to misunderstandings. I suggest that the author explain this in the manuscript or change the nomenclature of the GDL.
- 4- Why do the authors use a Nafion (Cation exchange) membrane when the reaction takes place in alkaline media? An anion exchange membrane would be more appropriated.
- 5- Line 100 and 101: the authors should mention the current density for the FE of 94.6%.
- 6- The paragraph starting on line 123 is irrelevant (specially the use of nano and microparticles) as the main factor that leads to higher currents is the increase of the surface area. This should be reduced in the main text and transferred to the supporting information.
- 7- Line 168 to 170: A reference should be added to this sentence.
- 8- Results and discussion related with the reaction mechanism (line 200). These experiments and results only show the pathway for the formation of diethylamine and not the reaction mechanism

for the electroreduction of acetonitrile to ethylamine. I think this is unnecessary for the manuscript since the authors claim high selectivity for ethylamine.

POINT-BY-POINT RESPONSE TO REVIEWER COMMENTS

Reviewer #1 (Remarks to the Author):

This article presents a good and potentially highly applicable results, the authors are absolutely correct that using electrochemistry to produce organic chemicals and feedstocks could be a great way to use at times excessive solar, wind, and other renewable energy. The study is well executed at a good level, shows a good understanding of the chemistry in question and is potentially very high impact.

However, they need to review and cite the literature more carefully before this can be published. In what way do they mean it is the "first-of-its-kind study of electrochemical reduction of acetonitrile to ethylamine"? Because there is definitely some prior art. For example, Dalton Trans 2019, 48, 9576 presents an example with a molecular cobalt catalyst - much less promising in FE and stability than the current work, but nevertheless a study of the process in question. And there is older work based on solid-state metal catalysts - see for example Batanero et al, J Org Chem 2002, 67, 2369; Foley et al, Can J Chem 1988, 66, 201 - and I think there are other examples going further back than this.

In summary, this is potentially publishable in Nature Comms but it needs to properly acknowledge prior work, and in the context of that work - largely published in much lower impact journals - make a convincing case for the results and insights obtained.

Reply: We thank the reviewer for the constructive suggestions. In current study, we for the first time demonstrated the electroreduction of acetonitrile to ethylamine with a high selectivity (Faradaic efficiency) and good stability under industrially relevant current densities. We acknowledge that nitrile electrochemical reduction phenomena have been reported in the literature and apologize for the confusion. We modified the manuscript accordingly.

The following literature survey has been added in the updated manuscript to address reviewer's comments:

".....Activation of acetonitrile electrochemically has been reported in the literature. Previous studies showed that acetonitrile can be converted to 3-aminocrotonitrile anion on a platinum electrode in non-aqueous electrolyte^{1, 2}. Additionally, the chemisorption of acetonitrile on Pt electrode coupled with two consecutive electron and proton transfer processes was qualitatively investigated using cyclic voltammetry, in situ infrared spectroscopy, and online mass spectrometry^{3, 4, 5, 6, 7}. More recently, Child et al., utilized homogeneous cobalt-based molecular catalysts to electrochemically reduce acetonitrile to ethylamine using acetic acid as hydrogen source in organic electrolyte⁸. Despite of these efforts, electrochemical reduction of acetonitrile suffers from low faradic efficiency (~22%) and low current density (~0.04 mA cm⁻²) and has not been systematically investigated using metallic electrocatalysts. "

References:

1. Foley JK, Korzeniewski C, Pons S. Anodic and cathodic reactions in acetonitrile/tetra-n-butylammonium tetrafluoroborate: an electrochemical and infrared spectroelectrochemical study. *Can J Chem* **66**, 201-206 (1988).
2. Batanero B, Barba F, Martín A. Preparation of 2,6-Dimethyl-4-arylpyridine- 3,5-dicarbonitrile: A Paired Electrosynthesis. *The Journal of Organic Chemistry* **67**, 2369-2371 (2002).
3. Angerstein-Kozłowska H, MacDougall B, Conway B. Electrochemisorption and reactivity of nitriles at platinum electrodes and the anodic H desorption effect. *Journal of Electroanalytical Chemistry and Interfacial Electrochemistry* **39**, 287-313 (1972).
4. Szklarczyk M, Sobkowski J. The behaviour of high polar organic solvents at platinum electrode—II. Adsorption and electrode reactions of acetonitrile. *Electrochimica Acta* **25**, 1597-1601 (1980).
5. Wasmus S, Vielstich W. Electro-oxidation and electroreduction of acetonitrile in aqueous acid solution: A DEMS study. *J Electroanal Chem* **345**, 323-335 (1993).
6. Morin S, Conway BE, Edens GJ, Weaver MJ. The reactive chemisorption of acetonitrile on Pt(111) and Pt(100) electrodes as examined by in situ infrared spectroscopy. *J Electroanal Chem* **421**, 213-220 (1997).
7. Reshetyenko TV, St-Pierre J. Study of the acetonitrile poisoning of platinum cathodes on proton exchange membrane fuel cell spatial performance using a segmented cell system. *J Power Sources* **293**, 929-940 (2015).
8. Child SN, *et al.* Cobalt-based molecular electrocatalysis of nitrile reduction: evolving sustainability beyond hydrogen. *Dalton Transactions* **48**, 9576-9580 (2019).

Reviewer #2 (Remarks to the Author):

The manuscript describes the electrochemical reduction of acetonitrile to ethylamine. The authors correctly explain that there is a need to explore the application of electrochemical methods for the synthesis of commodity and/or high valuable products. The manuscript includes a very systematic study of catalyst and reaction parameters as well as computational studies to provide information about the reaction mechanism of acetonitrile electroreduction. The topic and this study are of high interest to the scientific community and can be published in Nature Communications. However, before publication the authors should clarify and/or adjust the following points:

Reply: We thank the reviewer for supporting this work and below is our response to the comments.

1- Can the authors elaborate on the source (and production process) of acetonitrile used as starting material for the synthesis of ethylamine? The sustainability of the production process is highly dependent on the source of the substrate and acetonitrile has problems of itself. It is of high importance to have clarity when suggesting alternative synthetic routes.

Reply: We agree that we should discuss the source of acetonitrile in the manuscript. Commercially, acetonitrile is mainly produced through the Sohio process (annual production

capacity over 5MT) as a principal byproduct from the ammoxidation of propylene to acrylonitrile in the following reaction scheme^{1,2,3}.

Currently, only a small fraction of acrylonitrile producers recover acetonitrile and most of acetonitrile is burned as fuels, which emits a significant amount of NO_x ². Compared with traditional polluted nitrile incineration, this work provides an alternative process to convert the acetonitrile byproduct to value-added ethylamine.

The following text has been added in the introduction section to clarify the acetonitrile source.

“...There are several approaches for ethylamine production, such as amination of alcohols, reductive amination of aldehydes or ketones, and hydrogenation of nitriles. Among those methods, direct hydrogenation to ethylamine has drawn special interest due to unique acetonitrile raw material source. Acetonitrile is mainly produced as a byproduct in acrylonitrile production through the Sohio process. Currently, only a small fraction of acrylonitrile producers recover acetonitrile and most of acetonitrile is burned as fuels, which emits a significant amount of NO_x . Acetonitrile reduction to ethylamine could provide a more environmentally friendly process converting excess acetonitrile manufacturing capacity to value-added ethylamine product.”

References:

1. Acrylonitrile. In: *Ullmann's Encyclopedia of Industrial Chemistry*).
2. Nitriles. In: *Ullmann's Encyclopedia of Industrial Chemistry*).
3. Karp EM, *et al.* Renewable acrylonitrile production. *Science* **358**, 1307-1310 (2017).

2- The redox equations for the electrochemical process should be provided as they help not expert reader to better understand the proposed process.

Reply: We added the following redox equations to the revised manuscript as the reviewer suggested.

“The reaction schemes for nitrile electroreduction are as follows:

3- The authors used a gas diffusion layer as support for the catalyst used. However, the reaction does not involve gas reactants. In this case the GDL, becomes a simple carbon-based support and the use of GDL nomenclature can lead to misunderstandings. I suggest that the author explain this in the manuscript or change the nomenclature of the GDL.

Reply: Sorry for the confusion. We replaced nomenclature of “GDL” to “porous carbon paper”

in the revised manuscript.

4- Why do the authors use a Nafion (Cation exchange) membrane when the reaction takes place in alkaline media? An anion exchange membrane would be more appropriated.

Reply: Ideally, an anion exchange membrane would be more suitable for the proposed process. At the early stage of this work, we conducted the acetonitrile reduction reaction using a commercial anion exchange membrane (FAA3-PK-350, Fumasep) and found that the FAA3 membrane degraded rapidly under the reaction conditions. Therefore, we chose to use the cation exchange membrane (Nafion), which is known to be more robust in harsh conditions.

The following context is inserted into the updated manuscript to clarify rationale of membrane selection.

“Nafion 211 membrane was utilized in this study for its relatively high stability in the presence of organics.”

5- Line 100 and 101: the authors should mention the current density for the FE of 94.6%.

Reply: The current density was 50 mA cm⁻² when 94.6% faradic efficiency was achieved. We modified the text accordingly.

The following text is inserted to the manuscript.

“Among all the monometallic catalysts, Cu showed the highest ethylamine FE of 94.6% with a total current density of 50 mA cm⁻².”

6- The paragraph starting on line 123 is irrelevant (specially the use of nano and microparticles) as the main factor that leads to higher currents is the increase of the surface area. This should be reduced in the main text and transferred to the supporting information.

Reply: We moved the related discussion to the support information and updated the manuscript accordingly.

7- Line 168 to 170: A reference should be added to this sentence.

Reply: The following two references have been added to the revised manuscript.

Varela, A. S.; Kroschel, M.; Reier, T.; Strasser, P., Controlling the selectivity of CO₂ electroreduction on copper: The effect of the electrolyte concentration and the importance of the local pH. Catalysis Today 2016, 260, 8-13.

Dinh, C.-T.; Burdyny, T.; Kibria, M. G.; Seifitokaldani, A.; Gabardo, C. M.; García de Arquer, F. P.; Kiani, A.; Edwards, J. P.; De Luna, P.; Bushuyev, O. S.; Zou, C.; Quintero-Bermudez,

R.; Pang, Y.; Sinton, D.; Sargent, E. H., *CO₂ electroreduction to ethylene via hydroxide-mediated copper catalysis at an abrupt interface*. *Science* 2018, 360 (6390), 783-787.

8- Results and discussion related with the reaction mechanism (line 200). These experiments and results only show the pathway for the formation of diethylamine and not the reaction mechanism for the electroreduction of acetonitrile to ethylamine. I think this is unnecessary for the manuscript since the authors claim high selectivity for ethylamine.

Reply: Thanks for the suggestions, we acknowledge that major focus of this study is to demonstrate high selectivity to ethylamine from acetonitrile through electrochemical reduction. Detailed discussion of traceable di- and tri- ethylamine side products is not directly linked to the major conclusion. In the revision, we moved the discussion to the support information. The following text is revised to address reviewer's comments.

“Product distribution of acetonitrile electroreduction. We employed in-situ flow electrolyzer mass spectrometry (FEMS) to probe the reaction intermediates/products distribution during the electrolysis. FEMS offered a time-resolved detection with high sensitivity at approximately 10 - 100 μm from the electrode surface. A schematic diagram of the FEMS configuration is shown in Supplementary Fig. 13. A scan of fragments with mass to electron ratio (m/z) ranging from 1 to 90 was initially performed at -0.5 V versus RHE (approximately 100 mA cm^{-2}). The background subtracted spectrum (Fig. 1a) showed mass fragments corresponded to water, acetonitrile, hydrogen, and ethylamine. It is shown that traceable amount of diethylamine was formed during acetonitrile electrochemical reduction and no triethylamine was detected on the electrode surface, while they are two major byproducts in thermal acetonitrile hydrogenation (see Supplementary Fig. 14 for more detailed description). This result confirmed the high selectivity of ethylamine in acetonitrile electroreduction. We further examined the onset potential of each product using FEMS coupled with linear sweep voltammetry (0 to -0.76 V, scan rate of -5 mV s^{-1}) in 1 M NaOH containing 8 wt.% acetonitrile. The fragment intensities recorded in mass spectrometer were synchronized with the potentiostat signal (Fig. 1c). Hydrogen, ethylamine, diethylamine, and triethylamine signals were recorded at m/z 2, 45, 58, and 86, respectively. As shown in Fig. 1b, the onset potential of hydrogen evolution was about -0.21 V versus RHE, ethylamine started at -0.23 V, diethylamine signal emerged at -0.32 V and no change of triethylamine signal was observed. The FEMS results clearly show that ethylamine was formed at a lower overpotential than diethylamine, which suggests the high selectivity in acetonitrile electroreduction rely on the lower overpotential of ethylamine formation.”

Fig. 1|The online electrochemical differential flow electrolyzer mass spectroscopy (FEMS) detection of the product during acetonitrile reduction on Cu in 8 wt.% acetonitrile in 1 M NaOH. (a) mass spectrometric signals recorded while scanning in a mass range of 0-90 amu at -0.5 V versus RHE. (b) linear sweep voltammetry ranging from 0 V to -0.76 V versus RHE (scan rate of -5 mV s⁻¹). The fragment intensity of hydrogen, ethylamine, diethylamine and triethylamine recorded in mass spectrometer were synchronized with the potentiostat signal.

REVIEWERS' COMMENTS

Reviewer #2 (Remarks to the Author):

The authors have properly addressed all the points raised by the referee's improving the quality of the manuscript. As mentioned before, the manuscript is of high importance and I recommend publication without further changes.